# Food and Nutrient Intake during Pregnancy in Relation to Maternal Characteristics: Results from the NICE Birth Cohort in Northern Sweden

**DOI:** 10.3390/nu11071680

**Published:** 2019-07-22

**Authors:** Mia Stråvik, Karin Jonsson, Olle Hartvigsson, Anna Sandin, Agnes E. Wold, Ann-Sofie Sandberg, Malin Barman

**Affiliations:** 1Department of Biology and Biological Engineering, Food and Nutrition Science, Chalmers University of Technology, 41296 Gothenburg, Sweden; 2Department of Clinical Sciences, Unit of Pediatrics, Umeå University, 90187 Umeå, Sweden; 3Institute of Biomedicine, Department of Infectious Diseases, University of Gothenburg, 41390 Gothenburg, Sweden

**Keywords:** nutrition, pregnancy, micronutrients, macronutrients, food intake, lifestyle, NICE study

## Abstract

Linkages between diet and other lifestyle factors may confound observational studies. We used cluster analysis to analyze how the intake of food and nutrients during pregnancy co-varies with lifestyle, clinical and demographic factors in 567 women who participated in the NICE (nutritional impact on immunological maturation during childhood in relation to the environment) birth-cohort in northern Sweden. A food frequency questionnaire, Meal-Q, was administered in pregnancy Week 34, and the reported food and nutrient intakes were related to maternal characteristics such as age, education, rural/town residence, parity, pre-pregnancy smoking, first-trimester BMI, allergy and hyperemesis. Two lifestyle-diet clusters were identified: (1) High level of education and higher age were related to one another, and associated with a diet rich in fruits, vegetables, whole grains and fish, and (2) smoking before pregnancy and higher BMI in early pregnancy were related to one another and associated with a diet that contained white bread, French fries, pizza, meat, soft drinks, candy and snacks. More than half of the women had lower-than-recommended daily intake levels of vitamin D, folate, selenium, and iodine. Complex lifestyle-diet interactions should be considered in observational studies that link diet and pregnancy outcome.

## 1. Introduction

While several observational studies have suggested that an adequate intake of micronutrients prevent several adverse pregnancy outcomes, apart from the preventive effect of folic acid supplementation on neural tube defects, randomized controlled studies of such factors are largely lacking [1]. From studies of populations of persons who suffer from nutrient deficiency, it is known that nutrients such as proteins, iron, zinc, selenium, iodine, folate, vitamin A, choline, and long-chain polyunsaturated fatty acids are important for fetal brain development and cognitive function [2,3]. Iodine is essential for the production of thyroid hormones, which, in turn, are necessary for normal growth and development [4].

Although multiple observational studies have suggested that a mixed diet that is rich in fruits and vegetables and that contains fish decreases the risk of preterm delivery [5] and gestational diabetes [6] and is optimal with respect to maternal weight gain and fetal growth [7], the dietary pattern is interlinked with many other lifestyle factors, including smoking, body mass index (BMI), education, age, and urban versus rural residency [5,8].

In the ongoing *Nutritional impact on Immunological maturation during Childhood in relation to the Environment* (NICE) birth-cohort, the ways in which exposure to different agents during pregnancy and in early life affects the subsequent health status of the infants are being explored. The primary aim of the NICE study is to investigate the effects of exposures to key environmental factors, such as nutrition, microbes, environmental toxins, and lifestyle, during both pregnancy and early childhood on the maturation of the infant’s immune system, as well as on the development of sensitization and allergy, infant growth, neurological development, and oral health [9]. A detailed examination of the maternal diet during pregnancy is an important and an essential component of the NICE cohort, as the maternal diet during pregnancy is associated with many different outcomes, both during pregnancy and during childhood. 

This paper presents details on the dietary intakes of various food items, as well as on the intakes of macro- and micro-nutrients by the pregnant mothers in the NICE cohort. In addition, we compare these intakes to predefined maternal characteristics (age, education, residential address, parity, pre-pregnancy smoking, early pregnancy BMI, allergy and hyperemesis), with the aims of: (i) Creating a detailed overview of the dietary habits in a cohort of pregnant women in northern Sweden, and (ii) examining the different maternal factors associated with the current dietary habits and food choices of pregnant women in Sweden. To do this, unsupervised hierarchical clustering was used to analyze the associations between maternal characteristics and food and nutrient intake. Hierarchical clustering is an excellent tool to visualize how food and dietary patterns correlate with different maternal characteristics as well as enables estimation of covarying factors within the two different variable sets. 

## 2. Subjects and Methods 

### 2.1. Study Design

The NICE study is an on-going, prospective, birth-cohort study conducted in Luleå, in northern Sweden, which had an inclusion period of February 2015 to March 2018 and planned follow-up until the child reaches four years of age. In total, 655 pregnant women were recruited. The inclusion criteria for participation in the NICE cohort were that the women were planning to give birth at the local hospital, i.e., Sunderby Hospital, and that they could communicate in Swedish. Information to all expecting parents living in the catchment area of Sunderby Hospital was given at their first visit to their local maternity clinic in gestation Weeks 10–12. Recruitment took place in conjunction with a routine ultrasound at the hospital around gestational Week 18. The NICE cohort is described in detail in the study protocol [9]. The study is conducted in accordance with the Helsinki Declaration and has been approved by the Regional Ethical Review Board in Umeå, Sweden (2013/18-31M, 2015-71-32). To be able to participate, all women had to sign a written consent. All participants have the right to withdraw from the study at any point and have their data removed from all study documentation. 

### 2.2. Dietary Assessment

The pregnant mothers received a web-based food frequency questionnaire (FFQ), called the Meal-Q, by e-mail during pregnancy Week 34, with questions regarding their diet during the previous month (meaning the diet during pregnancy week 31 to 34). Reminders were sent out by e-mail one and two weeks after the initial e-mail, and together with the final reminder, a text message was sent to their cell phones. The original Meal-Q was developed by researchers at the Karolinska Institute and has been validated against a web-based, seven-day weighted food diary [10,11]. Meal-Q has a meal-based and interactive format, i.e., only those who report consumption of a certain food will receive follow-up questions related to details of this consumption. 

Depending on the number of follow-up questions, the Meal-Q can comprise 102–174 questions. The follow-up questions collect information regarding the use of low-calorie products, fat quality, and consumed amounts of coffee, bread, sour milk, flaxseed, cheese, chocolate, candy, soup, liquor, supplements and/or probiotics. The original questionnaire was revised to collect information regarding fat quality, and the contents of heavy metals, sugar and probiotics. The modifications are listed in Appendix A. 

### 2.3. Food Intake

The intakes of different foodstuffs were reported in the Meal-Q as intake frequencies on a nine-point scale: ≥5 times/day, 4 times/day, 3 times/day, 2 times/day, 1 time/day, 5–6 times/week, 3–4 times/week, 1–2 times/week, and 1–3 times/month. In cases of no intake or intake less than once per month, the participants were instructed to leave the question blank. 

To be able to use the intake frequencies as continuous variables in the statistical calculations, all intake frequencies were converted to “times per week”. When the intake frequency was defined as an interval, e.g., for a frequency of 5–6 times/week, the intake frequency was set to 5.5. 

To compute different food groups, such as fruits and vegetables, fish, meat, dairy etc., the reported food intake frequencies were converted to intake in grams per day using the standard portions given by the Swedish National Food Database, which is managed by The National Food Agency [12]. Standard portions for three of the variables, flaxseed, chips, and popcorn, could not be found in the database. Regarding flaxseed, the Swedish population is advised not to exceed an intake of one to two tablespoons per day and, therefore, a normal portion was set at one tablespoon, corresponding to 8 g. For both popcorn and chips, normal portions were defined as 30 g, according to the information on the packages of the two leading brands. Regarding chocolate and candy, the participants were asked to specify the amount consumed on one intake occasion, and this value was used instead of the estimated normal portions. For information about the definitions of normal portions, see Appendix A.

### 2.4. Intake of Nutrients

The levels of intake of macro- and micro-nutrients were calculated in a Java-based software (MealCalc), which was developed and validated by the research group that created the Meal-Q, as previously described [10,11]. To estimate the amounts of nutrients and energy from each food product, normal portions were used (defined by the National Food Agency of Sweden) or based on pictures with specified amounts, as provided in the questionnaire. 

The Meal-Q also included questions about the intake of nutritional supplements, such as “Do you usually consume supplements containing vitamins, minerals or other supplements?” Women who answered “yes” or “yes, sometimes” were asked to specify the consumed nutrients, from a list, and how often they consumed them. Another question was with respect to the “Regular intake of other supplements”, selected from a list that included beta-carotene, selenium, zinc, magnesium and other supplements, such as Q10, Echinacea, ginseng, etc. Brand information was not included, and no information was collected regarding the contents of the supplements. Supplements were not included in the nutrition calculations.

As described in the validation studies [10,11], the Meal-Q underestimates the intakes of energy and macronutrients, as compared to a seven-day food record, with the total reported energy intake being 83% of that reported by the seven-day food record. To adjust for this underestimate, we calculated the relative intake of different nutrients as the intake (in grams/day) divided by the reported total energy intake (in MJ/day). These energy-adjusted intakes (in MJ/day) were then compared to energy-adjusted recommendations of nutrients intake for pregnant women, i.e., the recommended daily intake in grams per day (according to the Nordic Nutrition Recommendations [13]) divided by an approximation of a recommended intake of energy per day for pregnant women of 10 MJ/day. 

### 2.5. Inclusion and Exclusion Criteria for the Dietary Study

Participants who were included in the statistical analysis were those who completed the dietary questionnaire, had singleton pregnancies, and reported an energy intake that was within the range of 500–4000 kcal. For women (*N* = 18) who participated in the NICE cohort with two children, only the data from the first pregnancy were used. When the intakes of food and nutrients were compared between the two participating occasions for these mothers no significant differences were found for any food or nutrient intake (data not shown). 

### 2.6. Maternal Characteristics

The following maternal characteristics were examined in relation to food and nutrient intake: Age, education, residential address, parity (number of previous deliveries), pre-pregnancy smoking, early-pregnancy BMI (BMI at registration to maternity care, mostly occurring early in the first trimester), allergy and hyperemesis (see Table 1 for descriptive data on the characteristics). Age and BMI were analyzed as continuous variables. Education was categorized into three groups: (1) Nine years of schooling (elementary school), (2) 12 years of schooling (senior high school), and (3) more than 12 years of schooling (university studies or other education after high school). Residential address was used as reported in the questionnaire using one of following five alternatives: (1) Town (central), (2) town (suburb), (3) municipality in the countryside, (4) house in the countryside without animals or stable, and (5) house in the countryside with animals and stable. Information about parity was dichotomized into nulliparous or having delivered one or more children previously. Pre-pregnancy smoking was converted to ‘yes/no’, with daily smoking and irregular smoking being grouped together. Allergy and hyperemesis were registered in the medical charts by the midwives. 

The variables were chosen based on previous publications suggesting associations with maternal dietary intake [5,14,15,16,17,18,19,20]. Information regarding the educational level and residential address was collected in questionnaires sent out to the participants during pregnancy. Information about the other maternal characteristics was extracted from the medical charts in Norrbotten County and compiled into the SPSS format by a statistician. 

### 2.7. Data Analysis

For the data analysis, the IBM SPSS Version 25 (IBM, New York, NY, USA), R Version 3.5.1 and SIMCA Umetrics, Version 15.0.2 software packages were used. Correlation plots were constructed to depict correlations between intakes of different micro- and macro-nutrients and food items. The magnitude of the correlation is indicated by color in the plots, where red denotes a positive correlation, and blue indicates a negative correlation. The darker the color and the larger the dot, the stronger is the correlation. In addition, heat maps illustrating the relationships between reported dietary intake and maternal characteristics were created using Spearman’s correlation. Additionally, red denotes a positive correlation, and blue indicates a negative correlation. Both the correlation plots and heat maps are based on unsupervised, hierarchical, cluster analyses that automatically structure the variables and place correlated variables next to one another, so as to form clusters. Variables that showed correlations of >0.1 or <−0.1 were further tested with univariate methods to investigate the statistical significance of the association. Moreover, a principal component analysis (PCA) was performed that included maternal characteristics and food items. For dichotomous variables (parity, smoking, hyperemesis, and allergy), the Mann–Whitney *U*-test was used. For categorical variables (education and residential address), the Kruskal–Wallis test was used. For continuous variables (age and BMI), Spearman’s correlation test was used. The levels of statistical significance are denoted as follows: * *p* < 0.05, ** *p* < 0.01, *** *p* < 0.001. To take multiple testing into account, only *p*-values < 0.001 were considered as significant associations.

As described above, if a food item was consumed on less than one occasion a month the participants were asked to leave the question blank, which resulted in a missing value. Therefore, for these questions, a missing value was handled as if the responder actively had responded that they did not consume the food more than once a month, so this variable was set as zero for these subjects. 

## 3. Results

Of the 655 women included in the NICE cohort, 633 received the Meal-Q (Figure 1). Reasons for not receiving the questionnaire were: Inclusion in the study after pregnancy Week 34, delivery of the baby before gestational Week 34 (the survey was sent out during Week 34), intrauterine fetal death, or no valid e-mail address available. Of those who received the questionnaire, 597 (94%) responded to all the questions. Women who reported a daily energy intake of <500 kcal (*N* = 3) or >4000 kcal (*N* = 3) were excluded. In addition, women with multiple enrolments (*N* = 17 of the women with completed Meal-Q) were only included with their first pregnancy. This resulted in 567 women being included in the statistical analyses (Figure 1). 

### 3.1. Characteristics

The characteristics of the included women are shown in Table 1. Most of the included women were aged between 25 and 35 years and had more than 12 years of schooling. Half of the women were expecting their first child, and 63% of the participants lived in a city (Table 1). 

### 3.2. Food and Beverage Intakes

Table 2 summarizes the food items that the pregnant women reported that they had consumed in the previous months. The average reported intakes were: Meat, 160 g/day, fish, 40 g/day, vegetables, 145 g/day, fruits, 220 g/day, candy, 44 g/day, and snacks (popcorn, chips, ice cream, crackers, cakes, buns), 24 g/day. In total, eight women (1%) reported any consumption of alcoholic beverages during pregnancy. 

No data regarding vegan or lacto-ovo vegetarian food consumption were registered in the questionnaire. Instead, a proxy for vegan diet was created for women who reported that they consumed dairy, cheese, egg, fish, chicken, and meat less than once a month, which was the lowest frequency of consumption that it was possible to report. None of the women fulfilled the criteria for the vegan diet. A proxy for a lacto-ovo vegetarian diet was created for those women who reported consumption of fish, chicken, and meat less than once a month, and seven (1%) were assigned to this group. Complete avoidance of gluten was reported by 43 (8%) of the participants, and complete avoidance of lactose was reported by 103 (18%) of the participants. 

Breakfast, lunch, and dinner were reported to be consumed daily by 94%, 86%, and 97% of the women, respectively. Regarding eating habits, 23% reported that they often ate between meals, 11% reported that they often skipped meals, and 13% defined their eating pattern as irregular. Six women (1%) reported eating fast food >4 times/week. 

Figure 2 shows the correlations between the different food items. A dark-red color corresponds to a strong positive correlation, whereas a dark-blue color represents a strong negative correlation. The program clusters highly correlated variables and places them next to each other in the plot. One cluster (upper-left) was found to include fish, egg, fruits, cereals, and whole-grain products. Another cluster (middle) included whole-grain bread, cheese, coffee, pasta and rice with fiber, vegetarian dishes, vegetables, and salad as a meal. A third cluster (bottom-right) contained pizza, candy, white bread, French fries, and soft drinks. A small cluster that included game meat, milk, red meat, and meat products was also formed (Figure 2). 

### 3.3. Intake of Macronutrients

The median reported energy intake was 7240 kJ. Due to the underestimation linked to Meal-Q, the real intake was estimated to be 20% higher [11]. Intake of macronutrients is expressed as a fraction of the total energy intake, the “energy percent” (E%) (Table 3). Protein contributed 16 E% and carbohydrates 48 E%, of which dietary fiber contributed with two energy percentage points, corresponding to a median intake of 18 g/day. Fat contributed 36 E%, of which saturated fat contributed 15 E% and monounsaturated fat 13 E%. Polyunsaturated fat intake contributed in total 4.2 E%, of which omega-6 fatty acids contributed 3.2 E% and omega-3 fatty acids 0.8 E% (Table 3). Reported intakes in of fatty acids in grams per day can be found in Appendix A. 

### 3.4. Intake of Micronutrients

Table 4 lists the reported intakes of micronutrients, vitamins, and minerals expressed as energy-adjusted values (amounts/MJ). Reported intakes of nutrients in grams per day can be found in Appendix A. 

A correlation plot (Figure 3) shows how the intakes of macro- and micro-nutrients were related. Many different clusters of positively correlated variables were revealed, for example for different fatty acids. One cluster (middle part of the diagram) contained fiber, whole-grain, vitamin A, beta-carotene, fatty acid 18:3 (alfa-linolenic acid), vitamin E, and the sum of total polyunsaturated fat and fatty acid 18:2 (linoleic acid). Another cluster (further to the right) consisted of riboflavin, sugars, phosphorus, sodium, iodine, vitamin D, and vitamin B12. In addition, salt, magnesium, fatty acids 16:1 and 20:4 (arachidonic acid), and cholesterol formed a single cluster, and vitamin K, potassium, folate, and calcium formed another cluster (bottom right). 

Table 5 shows the intake of supplements. In all, 33% of the women reported daily intake of multivitamins, 39% had a daily intake of iron, 11% consumed vitamin D daily, and 10% consumed fish oil or omega-3 supplements daily (Table 5). 

### 3.5. Associations between Maternal Characteristics and Food Intakes

The association between food intake and maternal characteristics were visualized in a heat map (Figure 4). Similar to the correlation plots shown in Figure 2 and Figure 3, red color denotes a positive correlation, while blue color represents a negative correlation between the characteristics and the food item. To account for multiple testing, only comparisons showing *p*-values < 0.001 are discussed below. 

Higher age and higher level of education both correlated positively to the reported intake levels of fruits and vegetables. A high level of education also correlated positively to the reported intake levels of vegetarian dishes and correlated negatively to the consumption of white bread, while age correlated negatively to the intake of soft drinks. A high BMI in early pregnancy correlated negatively to the reported intakes of fruits, cereals, and yogurt. Smoking correlated negatively to the reported intake of fruits (Figure 4).

Having one or more previous children (parity equal to one or higher) was positively associated with the reported intake of pancakes. Having Swedish nationality was negatively associated with the reported intake of vegetables. Residential address (low value for living in the city center and higher value for a more urban residency) was positively correlated to the intake of game meat. Hyperemesis and allergy were not significantly associated with the reported intake of food items (Figure 4).

To examine further the associations between diet and lifestyle, a principal component analysis (Figure 5) was performed that included the maternal characteristics and food items. In the loading plot in Figure 5, maternal age and maternal education are located to the right along component 1, while smoking, and to some extent also BMI, residence, hyperemesis, and having a Swedish nationality are located to the left. Food items that had the strongest impact on this separation along component 1 were: Vegetables, fruits, whole-grain products, vegetarian dishes, salad as a meal, fish, probiotic drinks, and cheese (clustered together with high age and high level of education), and on the other side, white bread, candy, snacks, French fries, soft drinks, red meat, meat products, milk and pizza (clustered together with smoking and high BMI). 

### 3.6. Associations between Maternal Characteristics and Nutrient Intake

The heat map in Figure 6 shows the correlations between maternal characteristics and intakes of micronutrients and fatty acids. To account for multiple testing, only those differences that have *p*-values < 0.001 are discussed. Educational level was the factor that correlated most strongly with the intake of nutrients by the woman. Having a higher level of education was positively associated with higher intakes of folate, fiber, iron, vitamins C, B6 and E, potassium, monosaccharides, beta-carotene, linoleic acid, and total polyunsaturated fatty acids. Conversely, a lower level of education correlated with higher reported intakes of saturated fatty acids and palmitic acid (16:0) in particular, as well as a higher intake of disaccharides (Figure 6). Higher maternal age was associated with a similar nutrient intake pattern, in the same way as a high level of education was associated with higher intakes of fiber, vitamin A, and beta-carotene. Smoking before pregnancy was associated with almost the opposite nutrient intake pattern seen for a high level of education, albeit with a less-strong correlation, being significantly negatively associated with the consumptions of fiber, iron, folate, potassium, thiamine, vitamin B6, and the omega-3 polyunsaturated fatty acids EPA and DHA. Swedish nationality was negatively associated with the reported intakes of vitamin C and vitamin E. The BMI in early pregnancy correlated negatively to the reported intake of fiber with a significance level < 0.001, indicating that the higher the BMI the lower the fiber intake. In addition, higher BMI was significantly associated, with a *p*-value < 0.01, with higher intakes of total saturated fatty acids, the saturated fatty acids palmitic acid (16:0), stearic acid (18:0), protein and vitamin B12, whereas lower reported intakes of potassium and monosaccharides was significantly associated, with a *p*-value < 0.01, to higher BMI. Parity, residential address, hyperemesis, and allergy correlated only weakly, or not at all, to the reported intakes of micronutrients (Figure 6).

## 4. Discussion

The NICE cohort is a prospective, longitudinal birth cohort that follows pregnant mothers, the fathers/partners, and their children from pregnancy until four years of age with the aim of examining how exposure to different agents, such as diet, microbes, and toxicants, affects the pregnancy and outcomes for the child [9]. The women in the NICE cohort received the web-based, interactive, Meal-Q FFQ [10,11] in pregnancy Week 34. The women were instructed to report their eating habits over the past month. The aims of this study were to give a detailed overview of the dietary intake of specific food items and nutrients in the third trimester of pregnancy in these women, and to correlate the dietary intakes with the different maternal characteristics that have been suggested in the literature to be associated with food intake, i.e., age, education level, residential address, parity, pre-pregnancy smoking, first-trimester BMI, allergy, and hyperemesis. The results obtained from this study suggest that maternal age, education, BMI, and smoking habits are strongly correlated to food intake patterns. Thus, on the one hand, being older, being more highly educated, having a low BMI, and being a non-smoker are associated with intakes of fruit, vegetables, fish, whole-grain products, vegetarian dishes, salad as a meal, probiotic-supplemented drinks and cheese, and higher consumption of vitamins and minerals. On the other hand, having a higher BMI and reporting as having smoked before pregnancy are associated with the consumption of white bread, candy, snacks, French fries, soft drinks, red meat, meat products, milk, and pizza. 

The finding that a high level of education is associated with high reported intakes of fruits, vegetables, fish, vegetarian dishes, fiber, and vitamins is in line with the results of previous observational studies [17,21,22]. The level of education in the present cohort is extraordinarily high, with 98% of the women having more than the nine years of compulsory education and 71% having studied for more than 12 years. Similar to having a high level of education, being older is associated with eating more fruits and vegetables, as well as consuming larger quantities of many micronutrients. As expected, age and education are significantly positively associated, as suggested by both the unsupervised heat map model and the PCA model. Hence, one may think that some of the associations seen for age may be due to a higher level of education being linked to a more advanced age. In order to test this, linear regression models were performed with (a) age as the only predictor and (b) age and education as predictors for fruit and vegetable intake. The results did not change notably (B = 6.9, *p* < 0.001 and B = 5.5, *p* = 0.003, respectively, data not shown), suggesting that the associations found for age and fruit and vegetable intake are not an effect of increased educational level. Corresponding analyses were performed for education, with and without age included as a predictor, and neither did these results differ notably. Hence, the two predictors appear to have separate effects on fruit and vegetable intake, and likely other foods associated with a healthy diet. A similar association between age and education on the one hand, and a diet rich in vegetables and fruits on the other hand, was found also in the Swedish national survey Riksmaten-Vuxna 2010–11, in which a representative sample of 1797 women and men, in the age range of 18–80 years and living in Sweden, recorded everything that they ate and drank over four consecutive days in a web-based food diary [23]. 

Pre-pregnancy smoking and high BMI cluster together in the heat map. High BMI is associated with low intakes of fruit, yogurt, and cereals. The food intake patterns are reflected in the nutrient intakes, with low intakes of fiber and several minerals and nutrients and a high intake of saturated fat. This is in accordance with the results of the Norwegian Mother and Child Cohort Study, MoBa, in which pregnant women who had a higher BMI reported lower intakes of vegetables, fruits, nuts, whole-grain foods, and yogurt and higher intakes of snacks, chocolate and sweets, cakes and cookies, white bread, French fries, processed meat products, and sugar-sweetened drinks [5]. We show that having smoked, at least before pregnancy, is associated with a low intake of fruits. This is in accordance with the Riksmaten-Vuxna 2010–11 study [23], in which women who smoked reported low consumption of fruits and vegetables, dairy products, and cereals and lower intakes of energy, unsaturated fatty acids, folate, and iron [23]. 

In the loading plot from the PCA analysis, pre-pregnancy BMI clusters in the same corner as rural residence and Swedish nationality. However, BMI correlates neither to nationality nor to rural residence, when analyzed using the univariate methodology. Furthermore, in the heat maps, only rural residence and Swedish nationality cluster together. In another birth cohort, the FARMFLORA study, we have reported that pregnant women living on dairy farms have higher intakes of butter and full-fat dairy products and consume lower levels of margarine and oils, as compared to pregnant women living in the same rural areas but not on farms [8]. In the present cohort, very few of the women lived on farms. We show that rural residence is associated with higher reported consumption of different kinds of game meats, with game shooting being part of the traditional Swedish rural lifestyle, particularly in the northern part of the country. 

The strongest association with having previous children was found for reported intake of pancakes, which is a typical food enjoyed by children. The two maternal disorders analyzed, hyperemesis and allergy, cluster together but are not significantly associated with food intake. 

The Norwegian Mother and Child Cohort Study, which is a large population-based pregnancy cohort [5], identified the following three distinct dietary patterns among 66,000 pregnant mothers: Prudent, which is characterized by the consumption of vegetables, fruits, nuts, whole-grain foods, and yoghurt; western, characterized by the consumption of snacks, chocolate and sweets, cakes and cookies, white bread, French fries, processed meat products, and sugar-sweetened drinks; and traditional, characterized by the eating of boiled potato, fish, margarine, low-fat milk, and cooked vegetables. These three dietary patterns identified in the MoBa cohort were strongly associated with different maternal characteristics [5]. Having an older age was associated with eating more of the prudent and traditional diets and less of the western diet in the MoBa cohort. A similar result was seen for mothers with a higher level of education. Having a high BMI (>29) pre-pregnancy was positively associated with a western diet and negatively associated with a prudent diet; the same was found for smoking during pregnancy in the MoBa cohort [5]. Similar associations between the maternal characteristics and dietary intake patterns of pregnant women have been shown in other smaller observational studies in which pregnant women who have a higher pre-pregnancy BMI are younger, are less-well-educated, have more children, and are smokers or live in an urban area have lower intakes of fruit and vegetables and higher intakes of red and processed meats [5,14,15,16,17,18,19,20]. Similar complex lifestyle-diet interactions are shown here for the pregnant mothers in the NICE cohort, with a high level of education and higher age being related to a diet that is rich in fruits, vegetables, whole-grain foods, and fish, and with smoking before pregnancy and higher BMI in early pregnancy being associated with the consumption of white bread, French fries, pizza, meat, soft drinks, candy, and snacks. 

The calculated median energy intake reported in the Meal-Q by the women in the NICE cohort was 7.2 MJ (1740 kcal). The recommended daily energy intake for women with a sedentary lifestyle (which is assumed to characterize pregnant women in the third trimester) in this age group is around 8 MJ/day, to which another 2.3 MJ should be added to account for the increased demand for energy during pregnancy [13], giving a total of approximately 10 MJ/day. Thus, the reported intake of 7.2 MJ is only 70% of the estimated energy requirement. The Meal-Q has been validated with seven-day (weighed) food records [10,11]. In the validation study, the median energy intake was 7.7 MJ, which corresponded to 83% of the energy intake assessed with a seven-day food record [11]. It should be noted that the Meal-Q has not been validated for pregnant women. It may be that pregnant women under-consume energy due to emesis, fear of gaining weight or fear of consuming many foods that are on the “forbidden list”. Moreover, reporting dietary intake habits may be more difficult during pregnancy, since the dietary habits may change considerably compared to the non-pregnant period [24]. 

Most self-administered food frequency questionnaire underestimates the total intake. Ideally, they should be used to rank pregnant women according to low and high levels of intake of energy, nutrients, and foods, but not to assess the exact intakes in grams [25,26,27]. Therefore, the intakes of macro- and micro-nutrients reported in the form of ‘amount per day’ should be interpreted with caution. Relative values, i.e., intake reported as energy percent or energy-adjusted values, might prove to be more reliable. 

One can speculate that certain food items are more likely to be under-reported than others. It has been shown previously that it is more common to underestimate the intakes of sweets and snacks and to overestimate the intakes of fruits, vegetables, and fish [28,29], probably due to a wish to adhere to general norms and recommendations that discourage the former and encourage the latter. Since sweets and snacks are very energy-dense, underestimation of these food items has a strong impact on the estimated total energy intake.

Comparing energy-adjusted intakes of nutrients with energy-adjusted recommendations [13] shows that more than half of the women have intake levels that are lower than those recommended for vitamin D (median level of 0.85 µg/MJ/day vs. recommended level of 0.98 µg/MJ/day), iron (median level of 1.4 vs. recommended level of 1.5 mg/MJ/day), iodine (median level of 14 µg/MJ/day vs. recommended level of 17 µg/MJ/day), and selenium (median level of 5.5 µg/MJ/day vs. recommended level of 5.9 µg/MJ/day). However, supplement intake is not included in the nutritional calculations. Around one-third of the women ate multivitamins daily, while vitamin D supplements were consumed daily by 11% and iron by 39% of the women. Iron supplementation is usually prescribed at the maternity clinics in Sweden. Furthermore, 5% of the women took selenium supplements regularly. In the case of iodine, no specific supplements were registered, although iodine is added to some multivitamin preparations. Thus, the intake levels of the nutrients are presumably higher than what was calculated by us and reported here. For vitamin D, exposure to sunlight, which was not measured in this study, is a major source of vitamin D which has to be considered. The iodine intake is particularly difficult to examine, as one of the most important sources of dietary iodine is table salt that is supplemented with iodine. However, not all table salts are supplemented with iodine. No information is given in the Meal-Q about which salt, whether iodine-supplemented or not, the women use. Therefore, if the women have a low intake of iodine this needs to be confirmed with more specific questions about what type of salt they use. If the women are deficient in iodine, or the other nutrients showing a low intake, needs to be assessed with biological measures of nutritional status. This will be done in upcoming papers from the NICE-cohort. 

A limitation associated with this study is that the exact amounts of the specific food items that were consumed were not investigated, even though the results are presented in grams per day. There is a risk that the conversion of intake frequency into amounts does not fully reflect the real intake. Another limitation is that the food frequency questionnaire used has not been validated in pregnant women. 

A strength of the current study is the use of unsupervised hierarchical clustering analyses to examine how different maternal characteristics correlate with the intake of food and nutrients of pregnant women as well as with each other. Our paper visualizes the complex interaction between diet and lifestyle factors in a comprehensive way and consolidates the difficulties to ascribe outcomes such as BMI to the diet per se. Considering that dietary trend changes over time it is important to keep data on diet-lifestyle interactions up to date. The NICE-cohort was collected 2015–2018.

## 5. Conclusions

The dietary patterns of pregnant women who were living in northern Sweden in the period 2015–2018 and were included in the NICE cohort differ significantly in relation to several maternal characteristics. Higher age, a higher level of education, lower BMI, and being non-smoker are all associated with the reported intakes of fruit, vegetables, fish, and nuts, as well as higher consumption of vitamins and minerals. These complex lifestyle-diet interactions should be considered in observational studies that link diet and pregnancy outcome. As the dietary patterns are interlinked with other lifestyle factors, it is difficult to determine the specific significance of dietary intake in relation to pregnancy outcomes. More than half of the pregnant women in this cohort have reported estimated intakes of vitamin D, iron, folate, selenium, and iodine that are lower than the recommended levels. The nutritional status of these women will be investigated in upcoming papers. 

## Figures and Tables

**Figure 1 nutrients-11-01680-f001:**
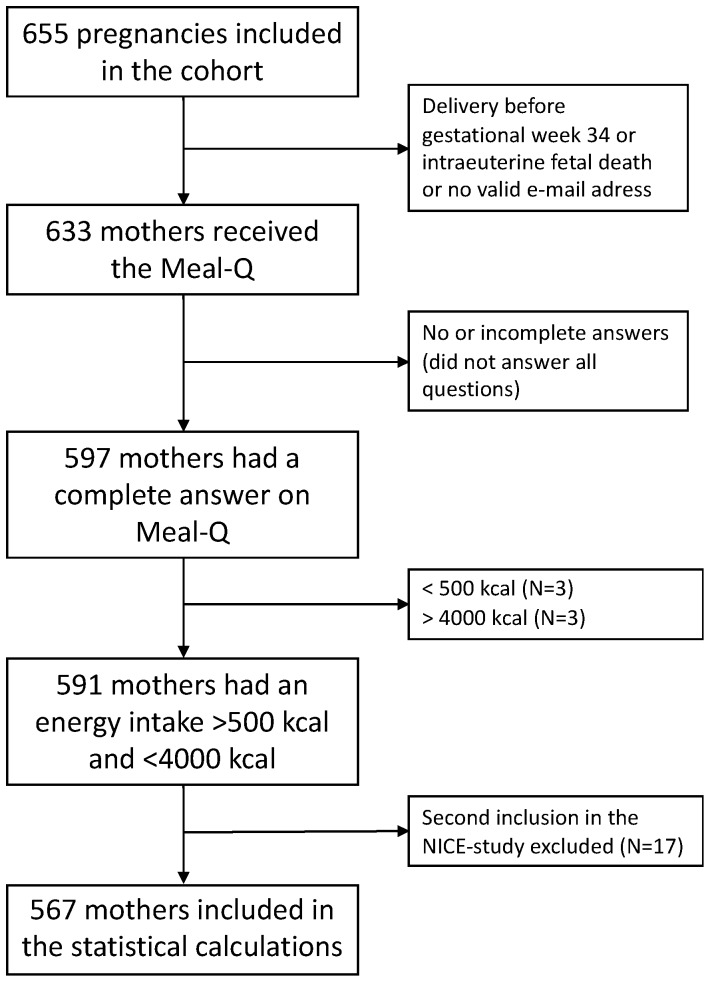
Flowchart regarding the inclusion and exclusion of subjects in the study.

**Figure 2 nutrients-11-01680-f002:**
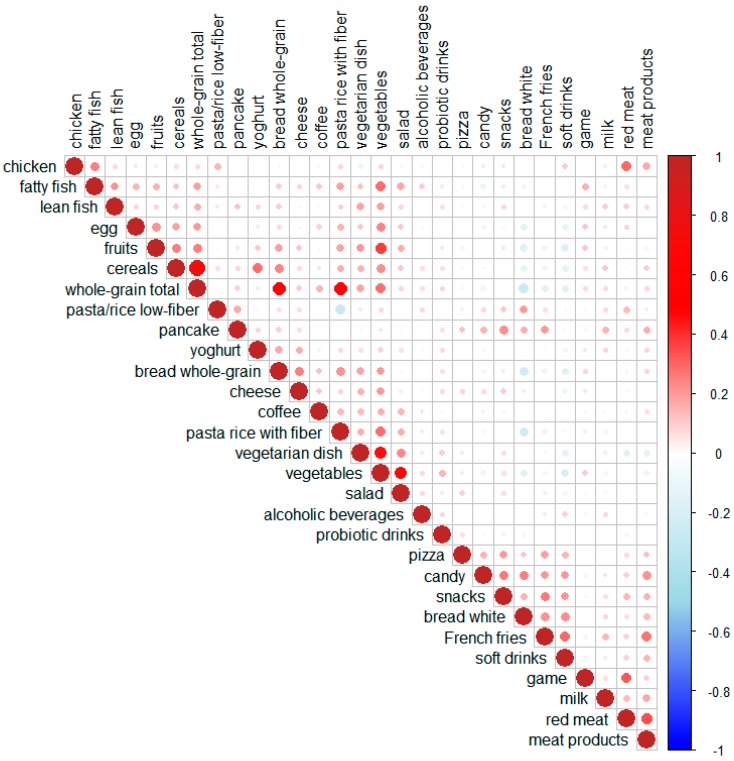
Correlation plot of food intake by all the subjects (*N* = 567). The model groups together all the variables that are highly correlated and places them next to one another. The magnitude of the correlation is denoted with a color, whereby red indicates a positive correlation and blue denotes a negative correlation. The stronger the correlation, the darker is the color, and the larger is the dot.

**Figure 3 nutrients-11-01680-f003:**
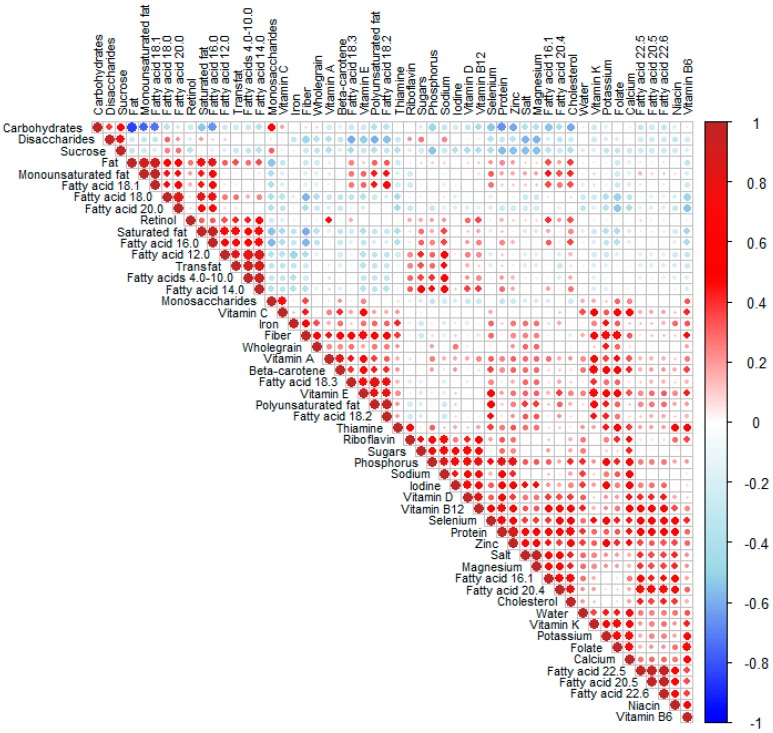
Correlation plot of energy-adjusted nutrient intake for all the subjects (*N* = 567). The model groups highly correlated variables and places them next to one another so that the correlated variables form clusters. The magnitude of the correlation is denoted with a color, whereby red indicates a positive correlation and blue indicates a negative correlation. The size of the correlation is reflected in the darker color. The larger size of the symbol indicates a stronger correlation, with lighter colors and smaller size showing weaker correlations, according to the scale on the right-hand side.

**Figure 4 nutrients-11-01680-f004:**
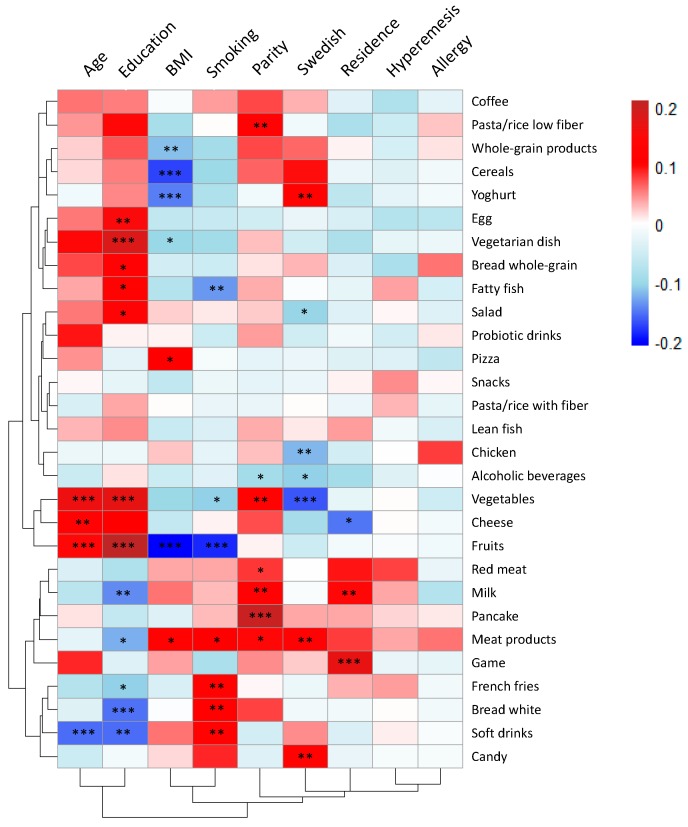
Heat map of correlations between food items and maternal characteristics (*N* = 567). The magnitude of each correlation is denoted with a color, whereby the red color indicates a positive correlation and dark-blue color represents a negative correlation., such that the deeper the color, the stronger is the correlation. Univariate statistics were applied to reveal significant associations. For dichotomous variables, i.e., parity, smoking, hyperemesis, and allergy, the Mann–Whitney *U*-test was used. For categorical variables, such as education and residential address, the Kruskal–Wallis test was used. For continuous variables, such as age and BMI, Spearman’s correlation test was used. Levels of statistical significance are denoted as: * *p* < 0.05, ** *p* < 0.01, *** *p* < 0.001.

**Figure 5 nutrients-11-01680-f005:**
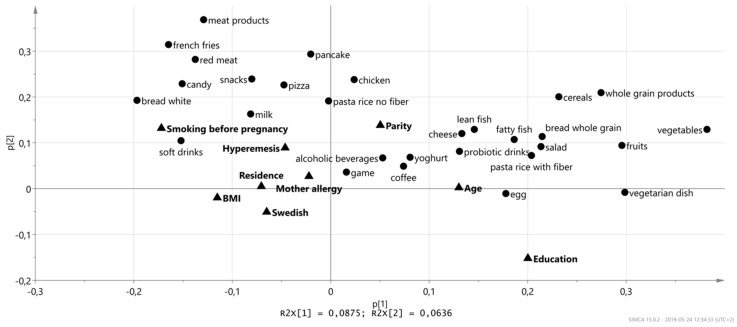
Loading plot from the principal component analysis (PCA) of selected food items and selected maternal characteristics. Food items that were found to be correlated to different maternal characteristics in the heat map models were aggregated with the maternal characteristics in a PCA model.

**Figure 6 nutrients-11-01680-f006:**
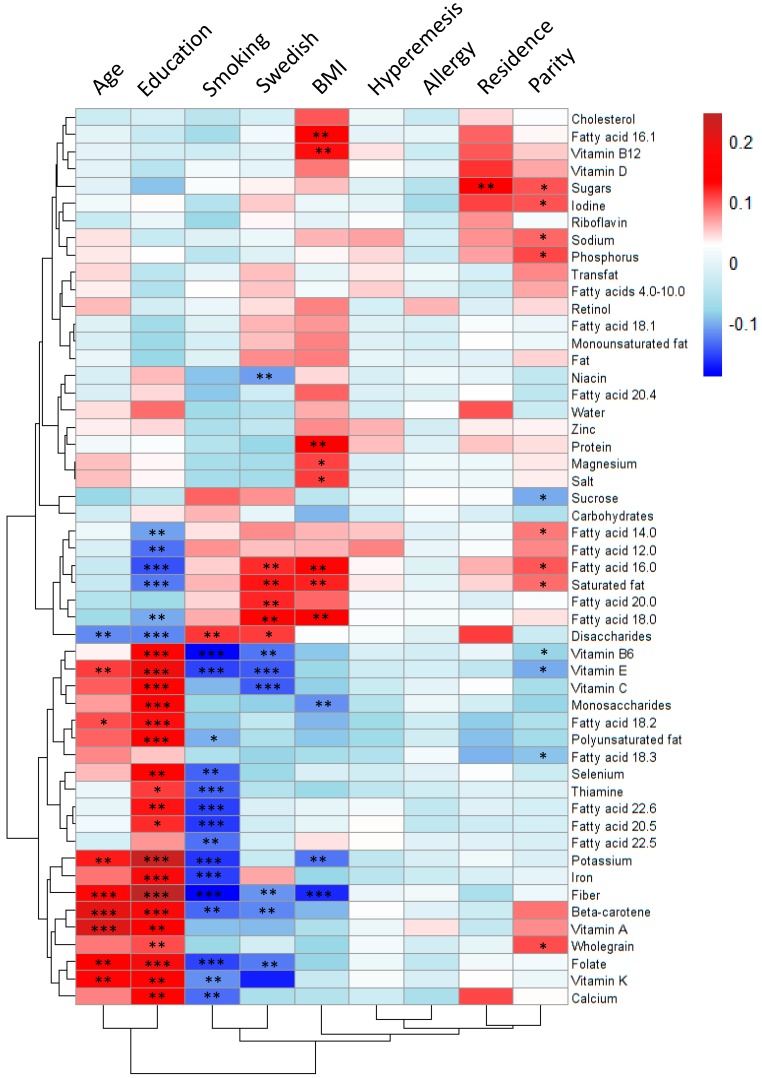
Heat map of correlations between micronutrients and maternal characteristics (*N* = 567). The magnitude of each correlation is designated by a color, whereby the dark-red color indicates a strong positive correlation, and dark-blue color represents a strong negative correlation. Univariate statistics were used to find significant associations. For dichotomous variables, i.e., parity, smoking, hyperemesis, and allergy, the Mann–Whitney *U*-test was used. For categorical variables, such as education and residential address, the Kruskal–Wallis test was used. For continuous variables, such as age and BMI, Spearman’s correlation test was used. Levels of statistical significance are denoted as: * *p* < 0.05, ** *p* < 0.01, *** *p* < 0.001.

**Table 1 nutrients-11-01680-t001:** Maternal characteristics (*N* = 567).

Maternal Characteristic	Number of Participants (%)
Age (years)	
≤25	71 (13)
26–30	235 (41)
31–35	172 (30)
≥35	89 (16)
BMI at registration to maternal care, kg/m^2^	
<18.5	9 (2)
18.5–24.9	293 (52)
25–29.9	155 (27)
30–34.9	56 (10)
≥35	26 (5)
Missing ^1^	28
Highest education level	
Elementary school (9 years of schooling)	12 (2)
Senior high school (12 years of schooling)	153 (27)
University or other education after high school (>12 years of full-time studies)	394 (71)
Missing ^1^	8
Smoking before pregnancy	
Yes	32 (6)
No	527 (94)
Missing ^1^	8
Parity	
No previous children	280 (50)
≥1 previous child	281 (50)
Missing ^1^	6
Marital status	
Married	23 (4)
Cohabitating	502 (94)
Other	7 (1)
Missing ^1^	35
Folic acid intake before pregnancy	
Yes	268 (49)
No	276 (51)
Missing ^1^	22
Residential address	
Town (central part)	88 (16)
Town (suburb)	256 (47)
Municipality in the countryside	65 (12)
Countryside without animals or stable	118 (22)
Countryside with animals and stable	13 (2)
Missing ^1^	27
Diseases	
Any allergy	211 (37)
Asthma	52 (10)
Bowel disease	6 (1)
Chronic hypertension	4 (1)
Diabetes type 1	3 (0.5)
Gestational diabetes	7 (1)
Hyperemesis	36 (7)
Heart disease	5 (1)
Rheumatoid arthritis	9 (2)

^1^ Not all women had information for all the variables due to missing information in the medical charts or due to the fact that not all the women completed the questionnaire that was distributed during pregnancy.

**Table 2 nutrients-11-01680-t002:** Dietary intake of food items, in grams per day (*N* = 567).

	Mean (SD)	Median (IQR)	Min–Max
Cereals ^1^	81 (119)	40 (20–99)	0–1384
Pasta/rice, low-fiber	74 (45)	75 (46–98)	0–260
Pasta/rice, with fiber	14 (22)	0 (0–33)	0–138
Bread, white	15 (17)	6 (2–30)	0–150
Bread, whole-grain	27 (25)	21 (9–40)	0–201
Bread, total	62 (45)	53 (35–78)	0–430
Whole–grain, total ^2^	100 (129)	64 (25–122)	0–1502
Milk	182 (219)	100 (14–200)	0–1000
Yoghurt	133 (130)	100 (43–200)	0–1000
Dairy products ^3^	344 (269)	266 (163–451)	0–1600
Cheese	13 (11)	11 (7–16)	0–75
Fresh fruits and berries	219 (159)	183 (105–297)	0–1246
Vegetables and roots	145 (90)	129 (84–185)	5–576
Total fruits and vegetables	363 (208)	322 (209–475)	11–1452
Red meat ^4^	66 (33)	66 (46–80)	0–214
Meat products ^5^	56 (32)	51 (36–75)	0–267
Game meat ^6^	12 (21)	0 (0–16)	0–125
Chicken	22 (16)	27 (9–27)	0–125
Total meat ^7^	157 (67)	153 (116–193)	0–469
Fatty fish	18 (14)	9 (9–28)	0–85
Lean fish	14 (12)	9 (9–27)	0–98
Seafood	5 (6)	7 (0–7)	0–50
Total fish	37 (23)	35 (18–50)	0–154
Vegetarian dishes	25 (44)	18 (0–23)	0–322
Egg	13 (17)	11 (4–11)	0–200
Coffee	148 (163)	118 (0–300)	0–750
French fries	16 (12)	11 (11–32)	0–75
Pizza	25 (22)	25 (25–25)	0–175
Nuts	7 (13)	3 (0–9)	0–120
Candy ^8^	44 (56)	29 (0–68)	0–400
Snacks ^9^	24 (19)	20 (10–34)	0–166

^1^ Includes seeds, oatmeal and cereals. ^2^ Includes different whole-grain products, such as pasta and bread. ^3^ Includes milk, chocolate milk, yogurt, sour milk, and cream. ^4^ Includes pork, beef, ground meat, bacon and lamb (not game meat). ^5^ Includes hamburger, sausage, black sausage, liver paste, and sandwich meat. ^6^ Includes meat from reindeer, elk, deer, wild boar and hare. ^7^ Includes red meat, meat products, game meat and chicken. ^8^ Includes candy and chocolate. ^9^ Includes popcorn, chips, ice cream, crackers, buns and cake.

**Table 3 nutrients-11-01680-t003:** Energy intake and intake of macronutrients, in energy percent (E%) (*N* = 567).

	Mean (SD)	Median (IQR)	Min–Max
Energy, kJ	7440 (2582)	7240 (5635–8888)	2421–16,441
Energy, kcal	1778 (617)	1730 (1345–2124)	579–3930
**Macronutrient**	**E%**		
Carbohydrates ^1^	48 (5.6)	48 (44–51)	27–66
Dietary fibers ^1^	2.2 (0.84)	2.1 (1.6–2.7)	0.55–5.4
Proteins ^1^	16 (2.7)	16 (15–18)	8.6–27
Total fat ^2^	36 (4.4)	36 (33–39)	20–51
Saturated fat ^2^	16 (2.8)	15 (14–17)	6.9–25
Monounsaturated fat ^2^	13 (1.8)	13 (11–14)	6.9–18
Polyunsaturated fat, sum ^2^	4.6 (1.5)	4.2 (3.5–5.2)	2.2–12
Omega-6 polyunsaturated fat ^2^	3.5 (1.2)	3.2 (2.6–4.1)	1.5–9.5
Omega-3 polyunsaturated fat ^2^	0.89 (0.34)	0.83 (0.67–1.0)	0.36–3.8

^1^ E% of carbohydrates and proteins were calculated as intakes in gram * 4 * 100/total energy intake in kcal. ^2^ E% of fat sources were calculated as intakes in gram* 9 * 100/total energy intake in kcal.

**Table 4 nutrients-11-01680-t004:** Dietary intakes of micronutrients per day, adjusted per MJ (*N* = 567).

Micronutrient	Median (IQR)	Min–Max
Vitamin A (µg)	90 (73–120)	40–340
Vitamin B6 (mg)	0.24 (0.20–0.27)	0.11–0.46
Vitamin B12 (µg)	0.69 (0.56–0.85)	0.13–1.8
Vitamin C (mg)	14 (9.4–19)	0.79–65
Vitamin D (µg) *	0.85 (0.66–1.2)	0.14–3.6
Vitamin E (mg)	1.1 (0.95–1.4)	0.53–3.6
Vitamin K (µg)	4.2 (3.2–5.5)	0.74–17
Calcium (g)	0.14 (0.12–0.17)	0.06–0.35
Folate (µg) *	42 (35–49)	15–105
Iodine (µg) *	14 (10–18)	3.4–70
Iron (mg) *	1.4 (1.2–1.7)	0.52–3.9
Magnesium (mg)	41 (37–46)	25–74
Niacin (mg)	2.2 (1.8–2.4)	0.81–4.8
Phosphorus (mg)	190 (170–210)	100–340
Potassium (g)	0.39 (0.34–0.45)	0.21–0.64
Riboflavin (mg)	0.23 (0.20–0.27)	0.11–0.62
Sodium (g)	0.31 (0.28–0.35)	0.13–0.63
Selenium (µg) *	5.5 (4.4–7.1)	2.1–17
Thiamin (mg)	0.18 (0.15–0.20)	0.74–17
Zinc (mg)	1.3 (1.2–1.4)	0.65–2.0

Data shown are energy-adjusted total intake per day, amount per MJ. An asterisk (*) indicates a median intake that is below the recommended intake for a Nordic population [13].

**Table 5 nutrients-11-01680-t005:** Numbers of study participants reporting intake of supplements.

	**Daily Intake, *N* (%)**	**Weekly Intake, *N* (%)**
Multivitamins with minerals	185 (33)	48 (9)
Vitamin A	23 (4)	2 (0)
Vitamin B	42 (7)	5 (1)
Vitamin C	31 (6)	14 (3)
Vitamin D	62 (11)	13 (2)
Vitamin E	23 (4)	3 (1)
Folic acid	104 (18)	22 (4)
Iron	220 (39)	75 (13)
Calcium	35 (6)	4 (1)
Fish oil/omega-3	54 (10)	15 (3)
	**Regular Intake, *N* (%)**	
Beta-carotene ^1^	18 (3)	
Selenium ^1^	30 (5)	
Zinc ^1^	40 (7)	
Magnesium ^1^	60 (11)	

^1^ For beta-carotene, selenium, zinc and magnesium, the question was “Do you regularly consume any of these supplements?”. No further question was asked regarding the frequency of intake of these supplements.

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
