# Peer review of "Food and Nutrient Intake during Pregnancy in Relation to Maternal Characteristics: Results from the NICE Birth Cohort in Northern Sweden"

_nutrients, 2019, doi:10.3390/nu11071680_

Round 1

Reviewer 1 Report

This is an interesting study that assesses the associations between maternal characteristics and nutrient intake during pregnancy. I think the study provides useful information regarding which maternal characteristics may need to be adjusted for or otherwise considered in studies examining associations between prenatal nutrition and health outcomes; however, it is not clear specifically what novel contribution this study makes to  the existing literature that has not previously been addressed. I suggest that the authors highlight this if the study is indeed addressing novel questions. 

Specific Comments

The authors state that the Meal-Q asks about diet during the previous “weeks”. Can they please specify the exact time frame that this FFQ asks about?

Line 128 – it may be helpful for the  authors to include a supplemental table comparing the nutrient intakes between the first and second pregnancies of the 18 woman who have two children rather than stating “data not shown”.

On line 149 it is stated that variables were chosen based on associations with maternal dietary intake in previous studies. If so, I suggest clarifying what this study is contributing  to the literature that is novel or that was not addressed in those studies.

Line  173 – I suggest incorporating “missing data” into the statistical analysis section rather than having it be its own section.

In Table 1 descriptive statistics are provided for “mental illness”; it would be helpful for the authors to describe what mental illnesses were assessed

Lines 330-335 – It is somewhat unclear what p values correspond with each of the BMI findings. I suggest re-writing this part for clarification.

The authors note in lines 370-372 of the discussion that the association between age and diet may be accounted for by higher education since age and education are correlated. Have the authors tested this in a regression model?

On line 407 the authors describe the “traditional” diet; however, margarine is not typically considered a traditional food. Conversely, it is more so an advent of the modern western diet. Why did the authors include this in the traditional diet?

In the discussion, I suggest the authors provide additionally commentary on the significance of the findings. That is, why are they important?

Minor Comments

Line 16 – I suggest changing “confound” to “may confound”

Line 25 – There should be a comma or semi-colon here instead of  a colon

Line 62 – I suggest changing “influence” to “are associated with” since this study cannot determine causality

Line 71 – I believe “were” should be changed to “was”

Line 322 – I suggest changing “low level” to “lower level”

Reviewer 2 Report

This is an interesting area of nutrition and an important one given what we know about the influence maternal nutrition appears to have on epigenetic alterations in offspring and the programming of gene expression via key metabolic pathways.etc. 

The results of the study presented here compared energy-adjusted intakes of various nutrients with energy-adjusted recommendations for example vitamin D (median level of 0.85 µg/MJ/day vs. recommended level of 0.98 µg/MJ/day). Given the fact that the best and most readily available sources of vitamin D are sunlight or supplements it is a shame these variables were not able to be considered and/or controlled for. However, the limitations of this study have been generally well presented by the authors. 

However, I would have liked to see some mention of the ethical procedures associated with this study i.e. signed consent etc. In addition, were women offered any follow up with dietetic services particularly if FFQ results highlighted nutritional safety issues? For example the small proportion of women who were excluded from the analysis due to implausible energy intakes?? Overall, well written and an important contribution to the literature and may be used for planning interventions with high risk groups
